# Rapid Identification of Beached Marine Plastics Pellets Using Laser-Induced Breakdown Spectroscopy: A Promising Tool for the Quantification of Coastal Pollution

**DOI:** 10.3390/s22186910

**Published:** 2022-09-13

**Authors:** Roberta Giugliano, Bruno Cocciaro, Francesco Poggialini, Stefano Legnaioli, Vincenzo Palleschi, Marina Locritani, Silvia Merlino

**Affiliations:** 1The Veterinary Medical Research Institute for Piedmont, Liguria and the Aosta Valley (IZS PLVA), U. O. Chimico, S. S. Sezione di Genova-Portualità, Piazza Borgo Pila 39/int. 24, 16129 Genova, Italy; 2Consiglio Nazionale delle Ricerche—Istituto di Chimica dei Composti Organo-Metallici (CNR-ICCOM), U. O. S. di Pisa, Area della Ricerca del CNR, Via G. Moruzzi, 1, 56124 Pisa, Italy; 3Istituto Nazionale di Geofisica e Vulcanologia, Sezione di Roma 2, Via di Vigna Murata 605, 00143 Roma, Italy; 4Consiglio Nazionale delle Ricerche—Istituto di Scienze Marine (CNR-ISMAR), U. O. S. di Pozzuolo di Lerici, c/o Forte Santa Teresa—Loc. Pozzuolo di Lerici, 19032 Lerici, Italy

**Keywords:** marine litter, laser-induced breakdown spectroscopy, resin pellets, environmental pollution, metals contamination

## Abstract

The rapid identification of beached marine micro-plastics is essential for the determination of the source of pollution and for planning the most effective strategies for remediation. In this paper, we present the results obtained by applying the laser-induced breakdown spectroscopy (LIBS) technique on a large sample of different kinds of plastics that can be found in a marine environment. The use of chemometric analytical tools allowed a rapid classification of the pellets with an accuracy greater than 80%. The LIBS spectrum and statistical tests proved their worth to quickly identify polymers, and in particular, to distinguish C-O from C-C backbone pellets, and PE from PP ones. In addition, the PCA analysis revealed a correlation between *appearance* (surface pellets roughness) and *color (yellowing)*, as reported by other recent studies. The preliminary results on the analysis of metals accumulated on the surface of the pellets are also reported. The implication of these results is discussed in view of the possibility of frequent monitoring of the marine plastic pollution on the seacoast.

## 1. Introduction

The occurrence of microplastics in the environment has attracted great attention as it has become a global concern. The most recent estimates put the amount of plastic reaching the oceans on a global scale at over 150 million tonnes, of which about 250,000 tonnes are fragmented into 5 trillion pieces that can float in water [1]. The economic damage caused to the fish market and tourism is estimated by the United Nations Environment Programme (UNEP) to be around 13 million USD each year [2]. The Mediterranean has one of the highest densities of waste, with around 62 million floating objects over the entire surface area, i.e., 17 times more than that estimated 35 years ago [3]. The origin of this huge amount of plastic is varied: 80% of the plastic comes from land and the remaining 20% from shipping [4,5]. Microplastics in a marine environment can originate from the photooxidation and fragmentation of macroplastics (“second generation microplastics”) or can be directly introduced into the marine environment (“first generation microplastics”). Among the latter, a high percentage in the microplastics range (from 1 to 5 mm) is represented by “resin pellets”. Pellets are granules of different polymeric types and are used to produce macroplastic objects via melting followed by extrusion and molding. Recent surveys have shown that they comprise about 30% of the microplastics present in the sea and on beaches (see, for example, Merlino et al. [6] and references therein). Biodegradable polymers (polylactide (PLA) and polybutylene adipate terephthalate (PBAT) have been recently added to the more commonly traceable pellets (mainly polyethylene (PE), polypropylene (PP) and polystyrene (PS)) [5] due to the commercialization and rapid diffusion of disposable products, which in recent years have been replacing the classic single-use products made of standard polymers, which were prohibited by the recent EU Directive n. 2019/904 and European Delegation Law 2019–2020.

It is feared that plastic can enter the food chain through voluntary or involuntary ingestion, causing damage not only through possible mechanical and physical effects due to their accumulation, but also through the chemical transfer of harmful substances from them to the stomachs of animals. These toxic substances can be absorbed from marine water or can be incorporated into plastics during industrial processing [7].

Recently, the problem of resin pellet pollution has created many concerns. Since there are still no strict regulations for the adoption of measures to prevent the possible loss of these millimetre plastics during their transport, storage and processing, the pellets are easily dispersed in the environment, and nowadays are present in many areas, including polar areas [8]. Field experiments have shown such pellets have a high capacity to absorb persistent organic pollutants (POP) [9,10] and metals [11].

Given the large number and variety of the pellets that can be recovered on sea coasts, it is important to develop methods for the rapid in-situ identification of the constituent materials of the pellets for individuating the origin of the plastic pollution but also for devising the best strategies for site remediation. Recent studies have shown how plastic pellets not only act as vehicles for metals in marine systems [12], but also act as an accumulator matrix [12,13,14]. An analysis of microplastics from marine organisms of the San Diego Bay area revealed a higher metal concentration (even by two orders of magnitude) than the one from the marine particulate matter of the same bay [13]. However, Rochman et al. [15] demonstrated that high-density polyethylene (HDPE) pellets typically accumulated lower concentrations of metals listed as priority pollutants by the US EPA (Cd, Ni, Zn and Pb) with respect to other kinds of plastics (polyethylene terephthalate (PET), polyvinyl chloride (PVC), low-density polyethylene (LDPE) and polypropylene (PP)). It has been shown that the microplastic pollution on the seacoast has a strong temporal and spatial dependence [6]. It is thus essential to develop fast and efficient methods for the analysis of microplastics capable of operating in situ. Because of their capability to absorb and concentrate pollutants from marine environments, over the years pellets have been used as non-living passive samplers to avoid the slow and high-cost preparation of samples of costal environmental media [16].

The techniques commonly used for the analysis of microplastics are FTIR and Raman spectroscopy (see Ref. [17] for a critical discussion). These techniques are complementary vibrational techniques and are often applied to analyse plastic debris extracted from sediments, water and living organisms, as described in Xu et al. [18]. Most of the FTIR and Raman analyses are preceded by manual visual sorting with a stereoscopic microscope, as described in Bruno et al. [19] and in the mentioned paper by Xu et al. [18].

These techniques, however, although generally precise in the determination of the pellets’ composition, are relatively slow and sensitive to the surface composition of the pellets. In addition, the Raman spectra technique exhibits strong fluorescence caused by micro-biological, organic and inorganic items on the plastic surface and the FTIR spectra interfere with water, therefore the pellets must be carefully dried. Moreover, no information about the accumulation of metals on the pellets’ surface can be obtained from the spectra. The hyperspectral imaging techniques recently described in a review by Faltynkova et al. [20], on the other hand, are in principle very fast, having the capability to analyze several pellets at the same time. The drawback of these systems is the high cost of the experimental apparatus required for the analysis (the ideal spectral range of analysis for plastics extends from 1000 to 2500 nm). As in the case of FTIR and Raman, hyperspectral imaging techniques do not give information about metallic pollution.

In this paper, we explored the possibility of setting up a new rapid strategy to monitor marine environments using the laser-induced breakdown spectroscopy (LIBS) technique. The advantages of LIBS with respect to the above-mentioned techniques area speed, ease of use and a relatively low cost of the experimental apparatus. Moreover, the LIBS technique is also capable of analysing the metal accumulation on the sample surface, thus assuring a full characterization of the pellets. 

## 2. Materials and Methods

To explore the feasibility of the classification of marine pellets by LIBS, we first analysed a set of 828 samples of virgin polymers (provided by Polymeric Materials Chemistry Laboratory of ICCOM-CNR), divided into 8 different classes corresponding to the polymers reported in Table 1.

For the analysis, we used the Modì mobile LIBS instrument [21], equipped with a dual pulse Nd:YAG laser, which emits two collinear laser pulses of about 20 ns FWHM at the wavelength of 1064 nm. The pulse energy was set to 30 mJ per pulse, with an interpulse delay of 1 µs. The acquisition delay was set to 1 µs (from the second pulse), with a gate of about 2 ms (time-integrated acquisition). The laser pulses were focused on the sample surface using a lens with 100 mm focal length to generate a plasma (the diameter of the sampled area is typically around 100 μm on the surface, with a depth of about 1 μm). The analysis takes place in the internal experimental chamber of the instrument. The LIBS signal was collected using an optical fiber and sent to an AvanSpec-USB2 spectrometer (from Avantes, NL) for acquisition. The LIBS spectrum, once acquired, was processed via the proprietary LIBS++ software. The analysis is very fast (fractions of seconds per sample) and minimally invasive, since a single laser shot is used for the analysis of the samples.

LIBS is essentially an elemental technique and, from an elemental point of view, the 8 polymers in Table 1 are very similar. Consequently, also the LIBS spectra of the samples are very similar. In Figure 1 we reported the average LIBS spectra of the eight polymers, showing almost no visible difference among the materials.

The prominent emission lines are, as expected, the carbon line at 247.9 nm, the CN violet molecular band between 370 and 390 nm and the hydrogen Balmer alpha line at 656.3 nm. Emission lines from Mg, Ca, Na and K are also visible in the spectra.

Given the strong similarity of the LIBS spectra, the classification of the plastic samples is not trivial. For differentiating the polymers, it is thus necessary to have recourse to chemometric methods, which have been demonstrated to be very efficient in similar studies [22,23,24,25,26,27,28,29,30,31,32].

Among the many classification methods available, we chose a simple artificial neural network classifier [33] (1 fully connected layer, 10 neurons) preceded by a principal component analysis (PCA) step for reducing the number of input variables. The LIBS spectra were zero-centered and normalized before the analysis. A 5-fold cross-validation was applied to test the classification capabilities of the model on unknown samples. All the calculations were performed using Matlab^®^ R2022a.

After this analysis, we studied a set of “marine” pellets, taken on different beaches on the coast between Tuscany and Liguria. We focused our attention specifically on PE and PP, since these polymers, due their density, can float on top of the sea’s surface and are predominant among the beached microplastics, and for these reasons are used as non-living passive samplers for marine pollution detection [16].

These pellets were collected at three beaches: “Spiaggia delle Grazie” (La Spezia), “Spiaggia di Marina di Pisa” (Pisa) and “Spiaggia del Calambrone” (Pisa), all three “natural” beaches, i.e., not subject to cleaning operations during the sampling period. 

The littorals under study do not present considerable daily tides that modify the extension of the beach. In any case, in order to avoid possible interferences, due to seasonality or variations in environmental conditions, on pellet accumulation rates, the collection of pellets was carried out over one week for all three beaches considered (March 2019). This choice is also corroborated by previous studies showing minimal seasonal variations in the density of pellets compared to other types of microplastics in the areas considered [6]. The number of pellets collected at each site (around 100) guarantees statistical significance, and the methodology used (zig-zag transects along a 100 metre stretch of beach, from the foreshore to the dunes or limiting structures in the inner part of the beach) is appropriate for our target, which does not include the study of the spatial and temporal variability of sample accumulation [34,35]. Pellets with a visible addition of dye (blue, black, brown, red, etc.) were discarded at the outset as they could not be used as a non-living passive samplers for the presence of additives added during manufacture [16,35]. All others (from transparent to white to yellow/amber) were placed in a glass container to avoid possible contamination with plastic and/or metal materials and then brought to the laboratory.

Then the samples were filtered with a Giuliani sieve (2 mm mesh), washed with deionized water and then placed in a dryer for two days. Once dried, pellets were classified by seven different parameters: *polymer type, dimension, shape, appearance, coloring, fouling* and the presence of *superficial spots* (see Figure 2).

Size, shape, appearance, fouling and presence of superficial stains were identified with a Dino-LITE optical microscope (AM4113ZT 10x–50x, 220x Polarizing Digital Microscope); meanwhile, for the classification of the colouring (yellowing), we followed the guidelines set out by Fanini’s work [35], which consisted of a comparison between the polymers with a colour scale used in the dental field [36]. *Size* was calculated by placing the pellets on a graph paper and observing them by optical microscope. *Appearance* consists of the presence or absence of surface roughness and porosity; pellets whose surface did not have porosity were classified as smooth, while pellets whose surface did not appear homogeneous were classified as porous. In other words, *appearance* parameter represents the pellets’ erosion caused by the permanence in sand or sea (oxidative degradation process) to the pellets’ surface. Pellets have been classified according to the presence or absence of fouling and surface stains. Fouling is a phenomenon due to the accumulation and deposition of living, animal and plant organisms, both unicellular and multicellular (biofouling), or other non-living substances, organic or inorganic [37]; *superficial spots* are biological matters but more are superficial and not inside plastic fractures and irregularities. 

Obtained data were analysed through Principal Component Analysis (PCA) [38]. The results of the physical classification of pellets were converted into numerical data according to the following nomenclature (Table 2).

## 3. Results

### 3.1. LIBS Classification of the Plastic Samples

The results of the analysis showed that a simple artificial neural network (ANN) classifier can guarantee a very high classification efficiency (the percentage of samples correctly classified is greater than 75%). The resulting confusion matrix is shown in Figure 3.

It can be observed that the true positive rate (true positives/total number) of PLA, PP and PVC is around or above 80%, with a maximum of 95.1% for PVC, while the true positive rates of PET, PBAT, NYLON, PS and HDPE are considerably lower. Notably, HDPE is mostly confused with NYLON (7.3%) and PLA (7.3%) and only at 3.7% with PP, while PP is, again, mostly confused with PLA (8.1%). Nylon is confused with HDPE (12.2%) and PBAT (11%), PBAT with NYLON (9.8%) and PET with PLA (12.2%) and PP (8.5%). Note that, to avoid possible bias in the classification due to the different number of samples per polymer type, we downsized the number of samples and performed the analysis on only 82 samples per type. 

Despite that, the classification efficiency of the model can be considered more than satisfactory, given that the LIBS spectra were taken with a mobile instrument. Moreover, just one spectrum per sample was acquired and all the spectra were used for the classification, including possible outliers resulting from laser intensity fluctuations and/or focusing issues on the transparent sample surface. 

### 3.2. Metal Analysis

As already described, “marine” beached pellets were analysed exploiting the same experimental setup used for the classification of the pristine pellets. As preliminary test, the LIBS spectra of a few of them were studied to detect possible contamination by metal pollution. We observed in the LIBS spectra (not shown here) weak emission lines of Al (394.4 nm and 396.2 nm), Fe (373.5 nm and 373.7 nm), Pb (405.8 nm) and Cr (425.4 nm and 427.5 nm). These lines were detected only in a thin layer corresponding to the portions of the surface covered by fouling or surface spots. The results are in accordance with the findings of Rochman et al. [15], which showed that the accumulation of metals does not occur directly on the polymer matrix but is mediated by the biofilm layer of marine fouling [39,40]. Further studies are planned to assess the feasibility of this kind of analysis; on the samples analysed, we verified that the lines of the metallic pollutants completely disappeared after the first laser shot on the surface. Therefore, in the analysis of beached pellets, it might be necessary to send two laser shots on the sample, the first to analyse the metals on the surface, and the second to determine the chemical nature of the sample. This can be completed in a fraction of a second, therefore the time of analysis would not be substantially affected.

### 3.3. PE and PP Resin Pellets Characterization

From the PCA analysis of the sub sample of PE and PP pellets collected in three Italian coastal areas (Figure 4), despite the relatively low variance (<50%) explained by the first two PCs, we observed a definite correlation between the *fouling* and presence of *surface stains* variables. The *appearance* and the *color* variables are also related to each other.

It is interesting to note that the size distribution of the pellets varies for the three beaches considered. On the ‘Le Grazie’ beach, we had about 30% of the pellets in class 1 (between 3 and 4 mm) and 70% in class 2 (between 4 and 5 mm). On the ‘Calambrone’ beach, the distribution changed to 40% in class 1 and 60% in class 2, while on the ‘Marina di Pisa’ beach, the pellets were equally distributed between class 1 and class 2. On all the beaches, the percentage of pellets in class 0 (between 2 and 3 mm) is negligible (<10%).

## 4. Discussion

This study highlights the possibility of easily and quickly distinguishing different polymer types among the samples collected during beach/sea microplastic surveys using LIBS.

So far, the golden standard technique to characterize pellets or plastic fragments is FTIR, but LIBS analysis is much faster (fractions of seconds for a single pellet). In addition, LIBS leads many advantages compared to FTIR, as it has the possibility to set up polymer characterization and metal contamination in the same analysis session (with the first ablation, the instrument detects metals on pellets surface, then with the second one, the laser ablation reaches the inner part of the polymer) and the option to collect information about the stratification of pollutants in the polymer matrix.

The scientific literature reports that PP and PE are the most common polymers used as marine pollution environmental proxies [16,35,41,42]. In fact, PE and PP, due their densities, can float on top of the sea’s surface, where most pollutants, especially polycyclic aromatic hydrocarbons (PAHs), are present. Other polymers, with higher densities, sink under the sea and are not easily collectable. In addition, following Endo et al. [34], prescriptions PE pellets have been used for a long time as passive samplers of POP pollution in the International Pellet Watch project [16,43,44], as they seem to present larger surface areas than PP and other common pellets, and to have an affinity for a wide range of organic contaminants varying in hydrophobicity [39,45,46,47]. The prescribed protocol for these studies involves the collection and separation of PE pellets from other types of collected beached pellets, with which they are easily confused if one relies solely on appearance.

Hence, it is therefore extremely useful to have a methodology that can quickly distinguish visible eroded and coloured PE pellets from other types of polymers, and particularly PP, considering that PE and PP together, due to their low specific gravity, account for about 80% of the beached pellets.

The PCA analysis conducted on our collected beached pellets revealed the correlation between the two pellet parameters of appearance and colour; surface porosity and staining formation are two parameters that are dependent on each other [48]. The highly oxidizing and stressful environment of the sea leads to the formation of cavities and therefore makes the surface porous [49]. This is because PE and PP degradation proceeds according to the Norrish II reaction [50]. The phenomenon is reported by other studies involving pellets collected from the beach and sea. They showed that prolonged residence in a marine environment produces an enhanced yellowing of resin pellets [51,52], which increases with time of exposure, as observed by Brandon in their long-term experiment conducted in tanks filled with seawater [36,53]. This has recently been confirmed by in-situ experiments on different types of pellets kept in sand and open water immersion [54], showing that a period of 6 months is sufficient to produce a visible change in colouration from white/transparent to yellow. However, the yellowing process can also be due to the absorption of pollutants by the surrounding environment: De Monte et al. [54] found that all sea samples show a more pronounced colour deviation towards yellow/amber compared to sand samples; this fact seems to indicate that in the first environment, i.e., the sea, the processes causing yellowing are enhanced.

The use of pellets as a “proxy” for pollution, widely used both for the International Pellets Watch and in subsequent studies, requires the analysis a large number of PE samples, which must be identified among all those sampled. The use of LIBS instead of the more classical method with the FT-IR makes it possible to greatly reduce the time required for pellet sorting and, at the same time, to obtain data on the metal pollution of the pellets, a type of study that is still scarce in the literature.

## 5. Conclusions

In this paper we have successfully tested the feasibility of an in-situ LIBS analysis of preproduction micro-plastic pellets, many of them collected from a marine environment.

While the cost of a portable LIBS instrument can be comparable to a portable FTIR or a Raman instrument, the main advantage of the technique is the possibility of performing a very fast analysis (fractions of seconds per pellet) on untreated samples, providing with the same measurement the classification of the pellet material and the concentration of the metals accumulated on the pellet’s surface. 

In fact, this study has demonstrated that the pellets can be classified with a good efficiency (>75% on the average) in a fraction of second, a figure that could increase substantially by applying spectral selection methods for outlier rejection. The LIBS spectrum and statistical tests proved valuable in quickly identifying polymers, in particular distinguishing C-O from C-C backbone pellets, and PE from PP ones. Moreover, a preliminary study has provided encouraging results on the possibility of detecting and measuring the metals accumulated on the surfaces of beached pellets. In addition, the PCA analysis on a sub sample of PP and PE pellets revealed a correlation between plastic *appearance* and *color*, as reported by other studies (De Monte et al. [54] and Endo et al. [34]). 

These findings could help monitor marine coasts in a more simple way and at a lower cost using easily collectable pellets, as proposed by Ogata et al. [16], who introduced the first International Pellet Watch (IPW).

## Figures and Tables

**Figure 1 sensors-22-06910-f001:**
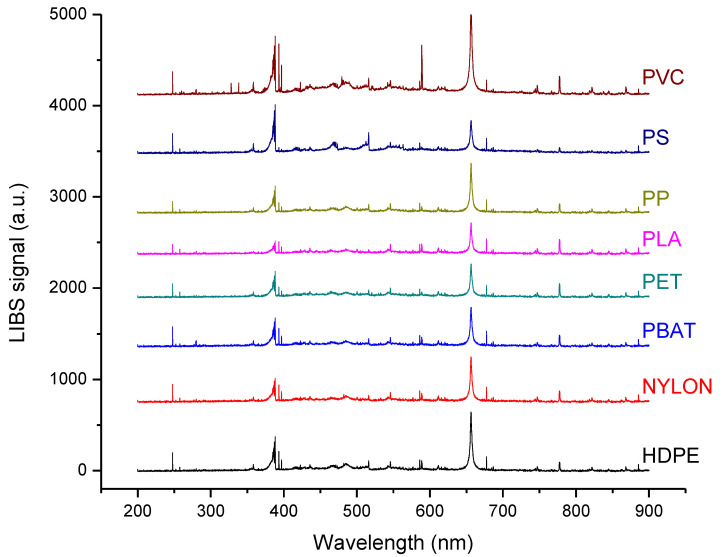
LIBS spectra of the plastic samples (average).

**Figure 2 sensors-22-06910-f002:**
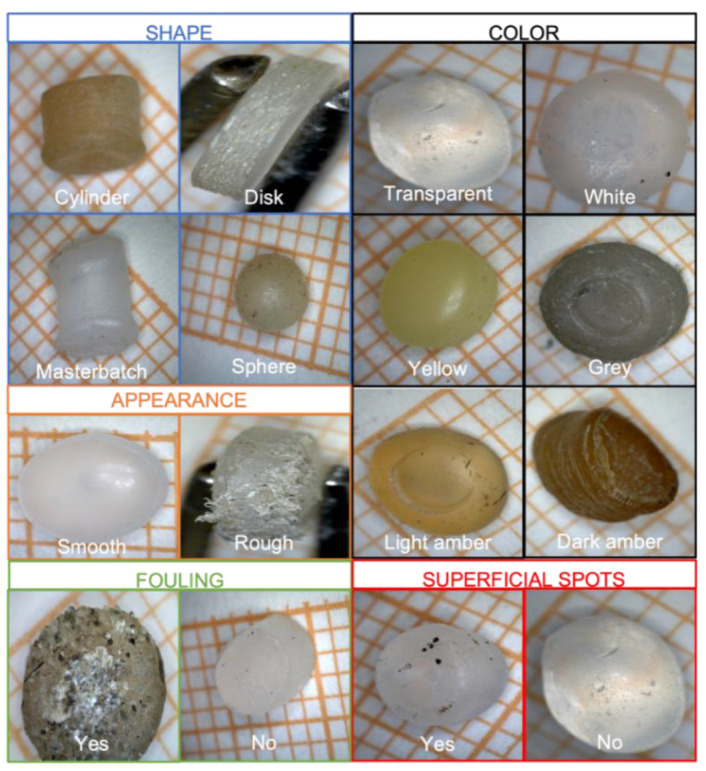
Different kinds of pellets. Masterbatch pellets are used in plastic manufacturing to carry pigments or other additives.

**Figure 3 sensors-22-06910-f003:**
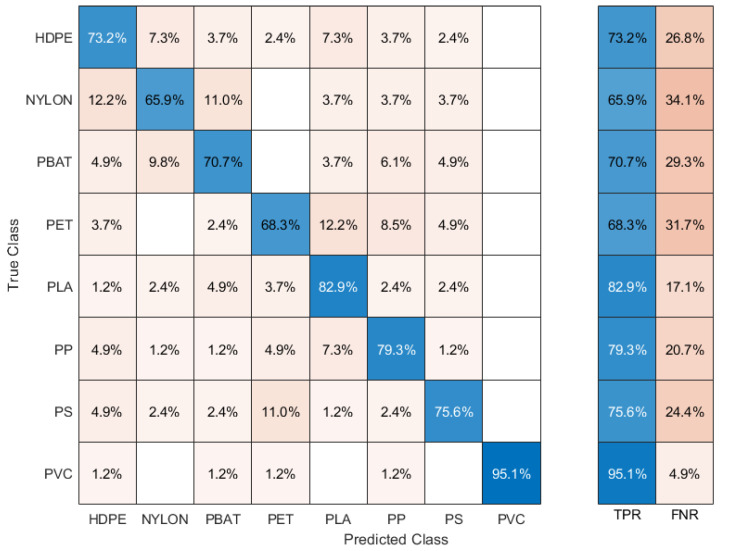
Confusion matrix resulting from ANN classification.

**Figure 4 sensors-22-06910-f004:**
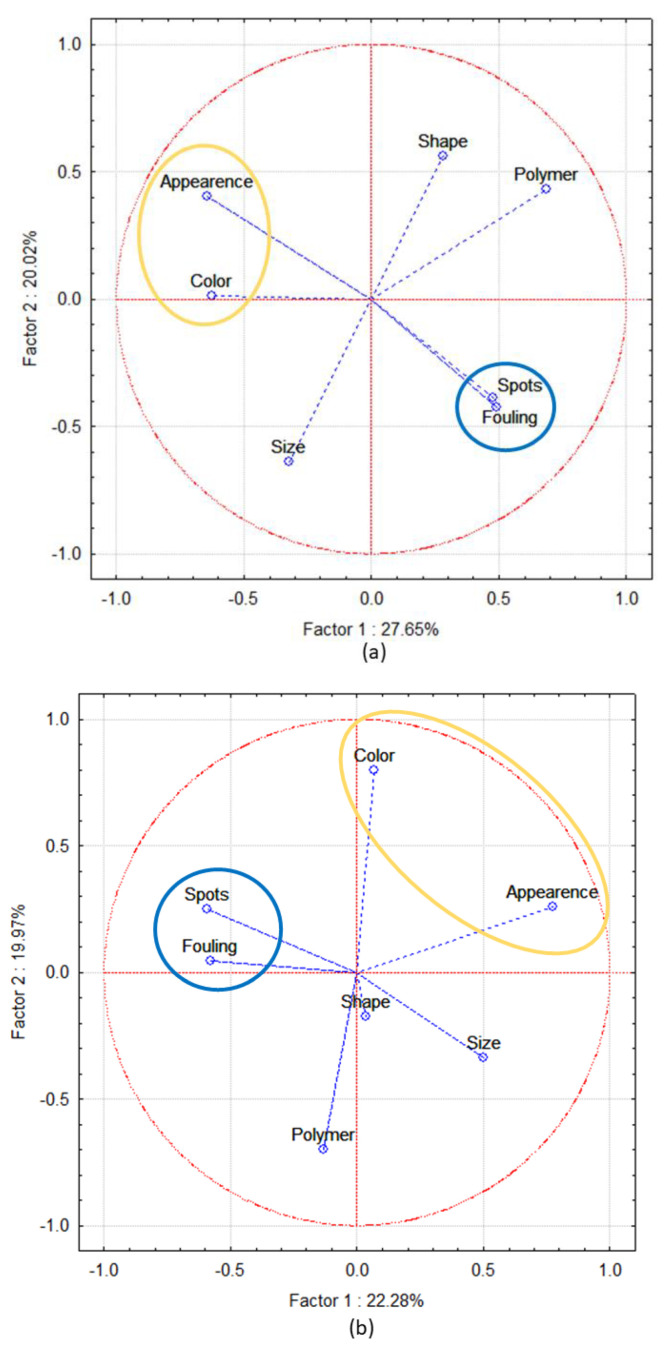
Principal component analysis (Factor 1 vs. Factor 2) of the pellets recovered from the beaches of (**a**) Le Grazie, (**b**) Calambrone and (**c**) Marina di Pisa.

**Table 1 sensors-22-06910-t001:** Materials and corresponding number of samples analysed by LIBS.

Material	Number of Samples Analyzed
HDPE (high-density polyethylene)	85
NYLON	148
PBAT (polybutylene adipate terephthalate)	99
PET (polyethylene terephthalate)	82
PLA (polylactide)	117
PP (polypropylene)	111
PS (polystyrene)	101
PVC (polyvinyl chloride)	85

**Table 2 sensors-22-06910-t002:** Parameters and numerical conversion for physical classification of the pellets.

Number	Parameter	Numerical Conversion
1	Polymer	PE → 0 PP →1
2	Size	2–3 mm → 0 3–4 mm → 1 4–5 mm → 2
3	Shape	Disk → 0 Cylinder → 1 Sphere → 2 Masterbatch → 3
4	Appearance	Smooth → 0 Rough → 1
5	Color	Transparent → 0 White → 1 Gray → 2 Yellow → 3 Light Amber → 4 Dark Amber → 5
6	Fouling	Yes → 0 No → 1
7	Superficial spots	Yes → 0 No → 1

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
