# Peer review of "Rapid Identification of Beached Marine Plastics Pellets Using Laser-Induced Breakdown Spectroscopy: A Promising Tool for the Quantification of Coastal Pollution"

_sensors, 2022, doi:10.3390/s22186910_

Round 1

Reviewer 1 Report

Paper: Rapid identification of beached marine plastics pellets using Laser-2 Induced Breakdown Spectroscopy: a promising tool for the quantification of coastal pollution.

Comments: In general, the paper describes a low coast tool of rapid identification of plastic pellets. I consider that paper must be improved to be accepted in the Sensors journal. The methodology section and results are not quite clear. There is a lack of information in sampling methods nor extraction of the plastics from the sand beaches matrix. It is not clear how the LIBS technique is capable of analyze metal accumulation. There are no images of the shapes nor classification of the pellets. The discussion and conclusion section is poor, must be improved. Can authors argue the differences in costs of the techniques?

In the manuscript there are acronyms not defined (i.e. line 54, 55, 205)

Line 65: A reference is needed to support the statement.

Line 66: what pellets problems?

Line 216: why the similar results are not shown in the paper?

Line 228: Why authors did not show the LIBS spectra lines?

Author Response

The methodology section and results are not quite clear. There is a lack of information in sampling methods nor extraction of the plastics from the sand beaches matrix.

The details on the sampling procedure are given in lines 170-181 of the revised manuscript.

It is not clear how the LIBS technique is capable of analyze metal accumulation.

The metals accumulated on the surface of the pellets are detected and eventually quantified from the characteristic emission lines of the elements.

There are no images of the shapes nor classification of the pellets.

We have added a figure (figure 2 in the revised manuscript) to show examples of the shapes and classification of the pellets.

The discussion and conclusion section is poor, must be improved. Can authors argue the differences in costs of the techniques?

We have clarified in the discussion and conclusions that, while the cost of a portable LIBS instrument is comparable to a FTIR or portable Raman instrument, the main advantage of the technique is the possibility of performing a very fast analysis (fractions of seconds per pellet) on untreated samples, providing with the same measurement the classification of the pellet material and the concentration of the metals accumulated on the pellet surface.

In the manuscript there are acronyms not defined (i.e. line 54, 55, 205)

We have expanded the acronyms in the text, as suggested by the reviewer.

Line 65: A reference is needed to support the statement.

The issue is discussed in ref. [7], we added the reference in the text.

Line 66: what pellets problems?

We apologize for not being clear in our statement, that have now been reformulated as: “Recently, the problem of resin pellet pollution has created many concerns.”

Line 216: why the similar results are not shown in the paper?

In the revised manuscript we have removed the sentence, since the classification obtained using different statistical tools was essentially identical to the results already shown.

Line 228: Why authors did not show the LIBS spectra lines?

The results on metal analysis are very preliminary, in the present configuration the lines detected are weak and cannot be used for a quantitative analysis. We are presently working in a systematic study aimed to improve the quality of the spectra; because of the complexity of the issue, we considered more appropriate not showing the corresponding LIBS spectra lines, leaving this topic to a future more complete discussion. 

Reviewer 2 Report

Dear Authors. I completly understand the motivations for the use of LIBS rather than FTIR and RAMAN. But, these arguments must be improved based in the following points:

- In the introduction section you made a quick argument about the use of FTIR and Raman, but everyone think about this techniques to characteriza polymers at first moment. The reference 17 was provided as decise on the topic, it is ok, but not enough - please provide 2 or 3 examples of studies thar use FTIR and/ou Raman for polymers classification associated with machine learning (multivariate analysis/chmometrics), and analyse/comment the results and conclusions provide more references! - also, discuss why LIBS is interesting in contrast.

- Provide more references some informations in the manuscript needs references.

- I recomment that you improve the Introduction and Materials and Methods section - too long and some informations are not necessary for the study.

- Remove Figure 1 there is no need to show a picture of the equipment.

- Improve the Figure and Captions of Fig 2 and 4. For figure 2 insert the name in the same collor of the spectra just above or bellow - remove from the box. Provide a better descrition in the captions. (why fig 2 is not in the results?). For Figure 4 remove the table around, identify as (A), (B)....and describe in the captions the meaning.

- The number of samples are not the same - how these can affect the ANN algorithm decision? I recommend a downsampling aproach and test the results again with the same number of samples in each class.

- I understood that the classification of Table II (numerical conversion) was used for PCA (Fig 4 results). So there is no practical apeal to use LIBS if I have to evaluate and classify the sample by using microscope. OR you just use the LIBS signal for PCA and as table II parameters? Please make it clear.

- 5 k-fold validation for number of sample as presented can create bias?

- Why you didn´t analyse certain spectral regions of LIBS? it could improve the overall accuracy - see the loading from PCA analysis.

Author Response

In the introduction section you made a quick argument about the use of FTIR and Raman, but everyone think about this techniques to characteriza polymers at first moment. The reference 17 was provided as decise on the topic, it is ok, but not enough - please provide 2 or 3 examples of studies thar use FTIR and/ou Raman for polymers classification associated with machine learning (multivariate analysis/chmometrics), and analyse/comment the results and conclusions provide more references! - also, discuss why LIBS is interesting in contrast.

Provide more references some informations in the manuscript needs references.

We have added several references, as suggested by the reviewer, on the specific use of FTIR and Raman analysis of microplastics.

I recomment that you improve the Introduction and Materials and Methods section - too long and some informations are not necessary for the study.

We have shortened the Introduction and Material and Methods sections, removing the information non necessary for the study.

Remove Figure 1 there is no need to show a picture of the equipment.

Improve the Figure and Captions of Fig 2 and 4. For figure 2 insert the name in the same collor of the spectra just above or bellow - remove from the box. Provide a better descrition in the captions. (why fig 2 is not in the results?). For Figure 4 remove the table around, identify as (A), (B)....and describe in the captions the meaning.

The figures have been modified following the reviewer’s suggestions.

The number of samples are not the same - how these can affect the ANN algorithm decision? I recommend a downsampling aproach and test the results again with the same number of samples in each class.

In principle, using a different number of samples for the different classes can produce a bias in the classification. Following the reviewer’s suggestions, we downsampled the number of pellets to have the same number in each class. The results obtained are substantially identical to the ones obtained without downsampling, but we agree with the reviewer that, from a methodological point of view, it’s appropriate to have the same number of samples in each class.

I understood that the classification of Table II (numerical conversion) was used for PCA (Fig 4 results). So there is no practical apeal to use LIBS if I have to evaluate and classify the sample by using microscope. OR you just use the LIBS signal for PCA and as table II parameters? Please make it clear.

There are two steps in the analytical workflow we suggested in the manuscript. With LIBS one can obtain the chemical information about the constituent polymer, while direct observation is used to study the correlation between the chemical (Polymer) and physical parameters (Shape, Size, Appearance, Color, Fouling and Spots).

5 k-fold validation for number of sample as presented can create bias?

For the number of samples considered a 5-fold cross-validation (80% of the samples used for building the classification model, and 20% of the samples used as external validation, with the procedure repeated five times) is considered appropriate for testing the classification capabilities of the model.

Why you didn´t analyse certain spectral regions of LIBS? it could improve the overall accuracy - see the loading from PCA analysis.

We used PCA for data reduction before optimizing the classifier, this is equivalent to remove from the LIBS spectra the less significant spectral features, retaining only the ones which are relevant for classification.

Reviewer 3 Report

In this paper, the authors have used LIBS measurements in order to study coastal pollution. The paper is quite clearly presented and the results could be of interest to the community.
I have some minor remarks:

1) line 73 ".. a rapid in-situ identification..", I do not completely agree with the need for in-situ measurement. Please explain clearly how an in-situ identification could affect the choice of a remediation strategy. Did you perform in-situ measurements with Modì? Unfortunately, in most cases, the strategy to follow is determined after a political rather than scientific course, hence I do not believe it is urgent to perform measurements on site.

2) line 104 "..relatively low cost..", just a curiosity, is MODI' relatively low cost? Please provide a comparison in terms of costs between a LIBS typical setup and FTIR or Raman ones.

3) line 128, there are two typoes e.g.

4) line 130 "..very fast.." please provide an estimate of how fast.

5) line 186 "size" since you have used a microscope to determine the size, could you please add a comment on the size distribution, it could be of interest to the reader.

6) line 187 "appearance" seems to be a qualitative feature, I suggest adding an image (or an inset) in order to understand how did you consider that a sample belongs to smooth or rough classes.

7) line 207 "..>75%.." How did you estimate this value? please add a comment.

Author Response

- line 73 ".. a rapid in-situ identification..", I do not completely agree with the need for in-situ measurement. Please explain clearly how an in-situ identification could affect the choice of a remediation strategy. Did you perform in-situ measurements with Modì? Unfortunately, in most cases, the strategy to follow is determined after a political rather than scientific course, hence I do not believe it is urgent to perform measurements on site.

We agree with the reviewer, however it should not be underestimated the possibility of having an instrument on-site for rapid assessment of the polymer type and metal accumulation. The information gathered in real time would be useful for optimizing the sampling strategy or when a rapid response (i.e. identification of the source of pellet pollution) would be needed. The measurements reported in the paper were performed in the laboratory, however the mobile LIBS instrument does not exploit any of the laboratory facilities, if not the power supply, therefore the same measurements could have been performed in situ. 

- line 104 "..relatively low cost..", just a curiosity, is MODI' relatively low cost? Please provide a comparison in terms of costs between a LIBS typical setup and FTIR or Raman ones.

The cost of a mobile/portable LIBS instrument is comparable to a FTIR or portable Raman instrument. We clarified in the revised manuscript that the main advantages with respect to other techniques are the short times of analysis and the possibility of detecting the surface metal contamination.

- line 128, there are two typoes e.g.

We thank the reviewer for pointing out the typos, we have corrected them in the revised version of our manuscript

- line 130 "..very fast.." please provide an estimate of how fast.

The analysis takes fractions of second per sample, we have added this information in the revised manuscript.

- line 186 "size" since you have used a microscope to determine the size, could you please add a comment on the size distribution, it could be of interest to the reader.

The size distribution of the pellets varies for the three beaches considered. In ‘Le Grazie’ beach we have about 30% of the pellets in class 1 (between 3 and 4 mm) and 70% in class 2 (between 4 and 5 mm). In ‘Calambrone’ beach the distribution changes to 40% in class 1 and 60% in class 2, while in ‘Marina di Pisa’ beach the pellets are equally distributed between class 1 and class 2. In all the beaches, the percentage of pellets in class 0 (between 2 and 3 mm) is negligible (< 10%). We have added the information in the revised manuscript.

- line 187 "appearance" seems to be a qualitative feature, I suggest adding an image (or an inset) in order to understand how did you consider that a sample belongs to smooth or rough classes.

We have added a figure (figure 2 in the revised manuscript) to show examples of the shapes and classification of the pellets.

- line 207 "..>75%.." How did you estimate this value? please add a comment.

The figure reported is the percentage of correctly classified samples. We have added this information in the revised manuscript.

Round 2

Reviewer 1 Report

The manuscript has improved substantially. All the reviewer observations have been attended properly. It would be really useful to finish up the development of a more power sensor that could analyze which elements nor metals could be present in the pellets surface, at least qualitatively. 

Reviewer 2 Report

No comments for the revised version.